# Simple Preparation of LaPO_4_:Ce, Tb Phosphors by an Ionic-Liquid-Driven Supported Liquid Membrane System

**DOI:** 10.3390/ijms20143424

**Published:** 2019-07-12

**Authors:** Jianguo Li, Hongying Dong, Fan Yang, Liangcheng Sun, Zhigang Zhao, Ruixi Bai, Hao Zhang

**Affiliations:** 1School of Chemical Engineering, Inner Mongolia University of Technology, Hohhot 010051, China; 2Key Laboratory of Design and Assembly of Functional Nanostructures, Fujian Provincial Key Laboratory of Nanomaterials, Fujian Institute of Research on the Structure of Matter, Chinese Academy of Sciences, Xiamen 361021, China; 3Xiamen Institute of Rare Earth Materials, Chinese Academy of Sciences, Xiamen 361021, China; 4Baotou Research Institute of Rare Earths, Baotou 014030, China

**Keywords:** LaPO_4_: Ce, Tb, ionic liquid, supported liquid membrane, photoluminescence

## Abstract

In this work, LaPO_4_:Ce, Tb phosphors were prepared by firing a LaPO_4_:Ce, Tb precipitate using an ionic-liquid-driven supported liquid membrane system. The entire system consisted of three parts: a mixed rare earth ion supply phase, a phosphate supply phase, and an ionic-liquid-driven supporting liquid membrane phase. This method showed the advantages of a high flux, high efficiency, and more controllable reaction process. The release rate of PO_4_^3−^ from the liquid film under different types of ionic liquid, the ratio of the rare earth ions in the precursor mixture, and the structure, morphology, and photoluminescence properties of LaPO_4_:Ce, Tb were investigated by inductively coupled plasma-atomic emission spectroscopy, X-ray diffraction, Raman spectra, scanning electron microscopy, and photoluminescence emission spectra methods. The results showed that a pure phase of lanthanum orthophosphate with a monoclinic structure can be formed. Due to differences in the anions in the rare earth supply phase, the prepared phosphors showed micro-spherical (when using rare earth sulfate as the raw material) and nanoscale stone-shape (when using rare earth nitrate as the raw material) morphologies. Moreover, the phosphors prepared by this method had good luminescent properties, reaching a maximum emission intensity under 277 nm excitation with a predominant green emission at 543 nm which corresponded to the ^5^D_4_-^7^F_5_ transition of Tb^3+^.

## 1. Introduction

Recently, rare-earth-ion-doped multicomponent compounds have attracted considerable attention due to their potential applications in the fields of electroluminescent devices, high-resolution displays, biological labels, and integrated optics [1,2,3,4]. Among these rare-earth-doped oxide phosphors, trivalent-cerium- and terbium-coactivated LaPO_4_ is significant because of its low solubility in water, its high thermal stability, and its high-efficiency energy transfer between Ce^3+^ and Tb^3+^ [5,6,7,8]. Due to the 4f orbital properties of La^3+^, lanthanide phosphate is transparent in the visible region and has been proven to be an ideal host structure for other lanthanide ions, resulting in luminescent materials in the UV-visible region [9,10,11]. In Ce^3+^ and Tb^3+^ co-doped LaPO_4_, Ce^3+^ with optically allowed d–f transitions is an effective activator for Tb^3+^ emission [12,13]. LaPO_4_:Ce^3+^, Tb^3+^ powders have been widely used as the green component of three band emission type fluorescent lamps [14,15]. In addition, LaPO_4_:Ce^3+^, Tb^3+^ phosphors have drawn continuous research attention in several other applications, including transparent fillers/markers, biomedical purposes, and plasma display panels [13,16,17,18].

Phosphor particles should be spherical in shape with no aggregation and their particle size should be in the micron range (<3 μm) with a narrow size distribution [16,19]. Spherical phosphor particles are more advantageous for the optical and geometric structure of the phosphor layer. The size of the phosphor affects the number of phosphor particles needed to produce the best coating for a particular application [20]. The shape, size distribution, and other microstructural characteristics of phosphors can be well controlled by different synthetic methods and reaction conditions. To date, several methods have been reported for the synthesis of phosphate phosphor materials, such as coprecipitation [21,22,23], solvothermal methods [24,25], electrospinning methods [26], solid-state methods [27], sol-gel processes [10,12], and spray pyrolysis [14]. Of these, coprecipitation is a common industrial synthetic method used to produce rare earth oxide powders and has the advantages of being feasible, low-cost, and environmentally friendly. Beyond that, fluorescent powders prepared by the coprecipitation method have uniform particle sizes, low agglomeration, and low phase impurities [21]. However, many factors such as reaction temperature, aging time, pH value, and solution concentration need to be controlled, which limits the development of coprecipitation methods. It is still challenging to simply prepare phosphors with favorable morphologies and excellent luminescent performance.

Ionic liquids are a kind of green solvent which includes a wide range of liquids, excellent thermal stability, a wide electrochemical window, and a low vapor pressure [28,29,30,31,32]. Recently, much attention has been paid to the study of ionic liquids in supported liquid membrane systems [33,34]. Ionic-liquid-driven supported liquid membrane systems have shown the advantages of high flux, high efficiency, strong durability, and environmental friendliness, and have made great progress in gas separation, organic separation, metal ion separation, and chemical reactions. Our team first committed to the use of an ionic-liquid-driven supported liquid membrane system to prepare CePO_4_ inorganic nanomaterials. In doing so, we could easily control the morphologies (rod or sphere) of rare earth luminescent materials by adjusting the pH and the concentration of SO_4_^2−^ [34]. Here, we used a facile ionic-liquid-driven supported liquid membrane method to prepare rare earth ion (Ce^3+^, Tb^3+^) co-doped LaPO_4_ phosphors with different morphologies (spherical and stone-like shapes). The preparation procedure, the role of the ionic liquid supported liquid membrane, characterization of the crystal structure, and photoluminescent properties of the synthesized LaPO_4_:Ce^3+^, Tb^3+^ phosphors are reported in the following sections. This method has been proven to be easily controlled, simple, and mild, and the phosphors prepared by this method show good morphological and photoluminescent properties.

## 2. Results and Discussion

### 2.1. Ionic-Liquid-Driven HVHP Membrane Characterization

In this experiment, we use a microporous ionic-liquid-driven HVHP membrane as selective ion channels that can selectively transfer PO_4_^3−^ from the PO_4_^3−^ supply phase into the mixed rare earth ion supply phase to prepare phosphors. We referred to the methods of Krzysztof A. et al. [35] to characterize the ionic-liquid-driven HVHP membrane. A Raman study was performed to investigate the ionic liquid presence on the surface and inside the HVHP membrane. The spectra of the inner part of the ionic-liquid-driven HVHP membrane was recorded up to 40 µm below the surface. As shown in Figure 1, the Raman vibration modes of ionic liquids can be observed on the surface of and in the interior of their corresponding functional membranes, which proves ionic liquids’ presence on the surface and inside the HVHP membranes. Figure 2 shows SEM micrographs of the cross-section of the untreated HVHP membrane, as well as SEM micrographs and the corresponding map microanalysis (B or S) of the cross-section of the resulting [C_4_mim][BF_4_]- and [C_4_mim][Tf_2_N]-driven HVHP membranes. The micrographs and map microanalysis show that the ionic liquid infiltrates the reticular surface of the membrane.

### 2.2. Membrane Reaction Mechanism

We referred to the mechanism study of PanPan Zhao et al. [34] to propose a possible membrane reaction mechanism. The entire process can be divided into two parts: the liquid membrane transport stage and the precipitation reaction stage. The porous HVHP membrane is a good hydrophobic barrier which can effectively separate the two aqueous phases. After being immersed in an ionic liquid, the function of the HVHP film changes significantly. The microporous HVHP membrane containing an ionic liquid consists of selective ion channels which can selectively transfer PO_4_^3−^ from the PO_4_^3−^ supply phase into the mixed rare earth ion supply phase; the microporous HVHP membrane without ionic liquid cannot do this (Figure 3a). A precipitation reaction occurs upon PO_4_^3−^ contacting the mixed rare earth ion supply phase on the other side of the HVHP membrane. Throughout the process, the cation (imidazolium) of the ionic liquid is responsible for the selective transfer of PO_4_^3^^−^ from the PO_4_^3^^−^ supply phase to the mixed rare earth ion supply phase. The anion is responsible for controlling the mixed rare earth ion supply phase and the release rate and ionic liquid hydrophobicity are correlated [34]. The hydrophilicity follows the order [N(SO_2_CF_3_)_2_]^−^ < [BF_4_]^−^, so the release rate of PO_4_^3^^−^ of the ionic-liquid-driven HVHP membrane is in this order (Figure 3a). The reaction appears to be a liquid-liquid extraction and occurs in the ionic liquid-film phase at the membrane interface. In addition, due to the thinness and high porosity of the porous HVHP membrane, the numerous ion transport channels are very short, meaning the precipitation reaction occurs quickly and efficiently. The experimental device and a schematic diagram of the reaction mechanisms are shown in Figure 4.

### 2.3. Transmittance of PO_4_^3−^ under the Action of the Two Functional Membranes

To investigate the transfer efficiency of PO_4_^3−^ by the different functional membranes, we performed the following experiment devices: two glass units sandwiching blank-, [C_4_mim][BF_4_]- or [C_4_mim][Tf_2_N]-infiltrated membranes. The glass units were filled with 50 mL deionized water and 50 mL of the phosphoric acid solution (1 M). To ensure a homogeneous system, both solutions were stirred with a magnetic stirrer at 1000 rpm. Samples of 10 µL were taken from the deionized water phase every 10 min. Then, the samples were diluted and the transfer of PO_4_^3−^ under the action of three kinds of membranes was measured with ICP. Figure 3a shows the changes of PO_4_^3−^ concentration in 50 mL of a deionized water phase under the action of three different membranes within 60 min. The PO_4_^3−^ concentration remained at 0 under the action of the blank-infiltrated membrane while the PO_4_^3−^ concentration increased under the action of the ionic liquid functional membranes. This suggests that PO_4_^3−^ cannot cross the blank-infiltrated membrane but can cross ionic liquid functional membranes. There is a clear difference between the [C_4_mim][BF_4_] functional membrane and the [C_4_mim][Tf_2_N] functional membrane, which indicates that the transfer efficiency of the [C_4_mim][BF_4_] functional membrane toward PO_4_^3−^ is much greater than that of the [C_4_mim][Tf_2_N] functional membrane. Figure 3b shows a picture of the phosphor reaction process and images of the precursor solution color when using rare earth sulfates as the rare earth supply phase for the different ionic liquid systems at different times under 254 nm irradiation. As shown in Figure 3b, precipitation occurs only in the rare earth phases, which indicates that the rare earth ions cannot pass through the membrane channels but phosphate can. The sample solutions all emitted green fluorescence when under a 254 nm light source and the fluorescence brightness increased with increasing reaction time. This proves that the membrane transfer rate of PO_4_^3−^ for the [C_4_mim][BF_4_] functional membrane is markedly faster than that for the [C_4_mim][Tf_2_N] functional membrane, which is consistent with the results of previous research. In the actual production application, we can choose an appropriate functional membrane with different ionic liquids to achieve the effect of controlling the rate of production.

### 2.4. Study of the Proportion of LaPO_4_:Ce^3+^, Tb^3+^

To explore the proportion of product, we dissolved an appropriate amount of the precursor (preparation conditions: 1 mol phosphoric acid solution, rare earth sulfate solution, and [C_4_mim][Tf_2_N] functional membrane) in a moderate amount of hydrochloric acid at 60 °C for 30 min, diluted the sample to the right concentration, and then tested the rare earth ion concentration by ICP. Table 1 shows the molar ratio of the mixed solution of rare earth elements from the rare earth supply phase and the molar ratio of the precursor. A clear difference between the solution proportion and precursor proportion can be seen in Table 1 because the rare earth ions do not completely precipitate. Then, we obtained the molar ratio (a fitting degree of greater than 99%) commonly used in the production of phosphate (La:Ce:Tb = 55:30:15) by simply adjusting the molar ratio of the rare earth ions in solution.

### 2.5. Structure and Morphologies of the LaPO_4_:Ce^3+^, Tb^3+^ Phosphors

In this study, we used an ionic-liquid-driven supported liquid membrane system to prepare phosphors. The whole system consisted of two glass units sandwiching a functional membrane ([C_4_mim][BF_4_] or [C_4_mim][Tf_2_N]). The glass units were filled with 50 mL of the rare earth mixture (rare earth sulfates or rare earth nitrates) and 50 mL of the phosphoric acid solution (1 M). The PO_4_^3^^−^ crossed the functional membrane to react with the rare earth ions in this system. Finally, the phosphors were prepared by calcining the precursors. The powder samples prepared from different rare earth ion sources (rare earth nitrates and rare earth sulfates) in the [C_4_mim][BF_4_] functional membrane were labelled BN and BS, respectively, and powder samples prepared from different rare earth ion sources (rare earth nitrates and rare earth sulfates) in the [C_4_mim][Tf_2_N] functional membrane were labelled NN and NS, respectively.

X-ray diffraction patterns were employed to determine the phase purities and crystal structures of the phosphor products. Figure 5a shows the XRD patterns of the precursors prepared under different conditions (different rare earth solutions and different ionic liquids). The vertical bars show the standard hexagonal LaPO_4_ peak positions (JCPDS No. 04-0635). Figure 5a shows that the diffraction peaks of all the precursors can be readily indexed to the hexagonal structure of LaPO_4_ in the P6222 space group (JCPDS No 04-0635). Figure 5b shows the XRD patterns of the as-prepared LaPO_4_:Ce, Tb phosphor samples prepared under different conditions (different rare earth solutions and different ionic liquids). The vertical bars show the standard monoclinic LaPO_4_ peak positions (JCPDS No. 32-0493). From Figure 5b, it is obvious that peaks at 2θ = 19.04°, 21.74°, 27.08°, 28.88°, and 42.48° are present after annealing at 1000 °C, which may be attributed to the (011), (101), (200), (120), and (221) reflections of the monazite crystalline structure of lanthanum phosphate. A monoclinic phase (space group: P21/n) of pure LaPO_4_ (JCPDS No. 32-0943) was obtained. By comparing the XRD pattern of the as-prepared precursors and LaPO_4_:Ce, Tb phosphor samples, we found that after annealing at 1000 °C, the fluorescent powder XRD peaks were sharper, the crystallinity was better and the structure of LaPO_4_:Ce, Tb had changed from a hexagonal to a monoclinic crystal phase. In addition, after the sample was calcined at 1000 °C, all the diffraction peaks shifted to the right compared with the standard diffraction peaks. This is because the radii of Ce^3+^ (~0.1034 nm) and Tb^3+^ (~0.0923 nm) are smaller than the radius of La^3+^ (~0.1061 nm) in the LaPO_4_:Ce^3+^, Tb^3+^ crystals, which leads to lattice contraction and a reduction of interplanar distance. Thus, based on the Bragg diffraction principle 2dsinθ = λ, where the decrease of the d value increases the diffraction angle, the diffraction peak positions of the XRD patterns move towards larger angles [23,36].

The morphology, size, and microstructural details were investigated by scanning electron microscopy. Figure 6 shows the SEM micrographs of the precursors prepared under different conditions (different rare earth solutions and different ionic liquids). Notably, the morphologies of the precursors are similar in the different ionic liquid functional membranes but show different morphologies for the different rare earth sources. When the anion of the rare earth mixed solution was sulfate, the samples exhibited a spherical morphology with particle sizes in the range of 600–800 nm, and a rough surface which consisted of aggregates of smaller particles. When the anion of the rare earth mixed solution was nitrate, the samples exhibited a flower-like structure with a diameter of approximately 30 nm and a length of approximately 200 nm. According to our previous research, we believe the reason for the formation of this globular structure is due to the template effect of SO_4_^2−^ [34]. Figure 7 shows SEM micrographs of the as-prepared LaPO_4_:Ce, Tb phosphor samples prepared under different conditions (different rare earth solutions and different ionic liquids). Similarly to the precursors, the annealed samples had similar morphologies when prepared with different ionic liquid functional membranes but different morphologies when prepared with different rare earth sources. After sintering, the samples prepared with rare earth sulfates as the raw material maintained their spherical structure, but the aggregation between spheres was more severe than that in the precursor samples. The small particles on the spherical surface of the sintered samples were larger than those on the surface of the respective precursor. However, after sintering, the samples prepared with rare earth nitrates as the raw material changed from a nanowire flower-like structure to a stone-like structure. The particle sizes were in the range 30–300 nm. The results show that the crystal size of all the samples increased after calcining, and due to the templating effect of SO_4_^2−^, the samples with rare earth sulfates as the raw material continued to maintain their large micro-sized spherical morphology, while the shape of the samples prepared using rare earth nitrates as the raw material grew from a flower-like structure into a stone-like morphology.

### 2.6. Photoluminescent Properties of the LaPO_4_:Ce^3+^, Tb^3+^ Phosphors

Figure 8 shows the excitation spectra of all the calcined LaPO_4_:Ce^3+^, Tb^3+^ phosphors by monitoring the ^5^D_4_→^7^F_5_ emission of Tb^3+^ (λ_em_ = 543 nm) at room temperature. As clearly shown in Figure 8, the obtained fluorescent materials absorb excitation energy in the range of 240–310 nm with a maximum excitation wavelength at 277 nm, which may be related to the f–d transitions of Ce^3+^. In addition, several small peaks can be detected in the range of 310–400 nm, which could be caused by the f–f transitions of Tb^3+^ [37,38]. Because of the forbidden nature of these transitions, their oscillator strength is much weaker than that of the spin-allowed 4f^1^–4f^0^5d^1^ Ce^3+^ transitions [22]. The excitation spectra consist of the strong excitation band of Ce^3+^ and the weak excitation bands of Tb^3+^, revealing that Tb^3+^ are essentially excited by Ce^3+^. In fact, several of the weak f–f excitation bands of Tb^3+^ are only present in the region of the Ce^3+^ emission. Thus, energy transfer from Ce^3+^ to Tb^3+^ occurs [21]. The emission spectra of all the calcined LaPO_4_:Ce^3+^, Tb^3+^ phosphors at an excitation of 277 nm are shown in Figure 9. All the calcined LaPO_4_:Ce^3+^, Tb^3+^ phosphors show obvious photoluminescence in the spectral range of 450–650 nm, and the four emission peaks at 487, 543, 584, and 621 nm can be assigned to the ^5^D_4_–^7^F_6_, ^5^D_4_–^7^F_5_, ^5^D_4_–^7^F_4_, and ^5^D_4_–^7^F_3_ transitions, respectively, of Tb^3+^ [39,40]. Among these peaks, the green emission at 543 nm, which corresponds to the ^5^D_4_–^7^F_5_ transition of Tb^3+^, is the predominant peak. The spectral properties of the phosphors prepared by an ionic-liquid-driven supported liquid membrane system are essentially the same as those prepared by other synthetic methods, in which the improved ionic-liquid-driven supported liquid membrane system is a new and effective method to prepare LaPO_4_:Ce^3+^, Tb^3+^ phosphors.

## 3. Materials and Methods

### 3.1. Materials

The lanthanum sulfate hydrate, cerium sulfate hydrate, terbium sulfate hydrate, lanthanum nitrate hydrate, cerium nitrate hydrate, and terbium nitrate hydrate were provided by the Baotou Research Institute of Rare Earths, and the phosphoric acid solution was purchased from Aladdin (Shanghai, China). The chemicals used in the experiments were of analytical grade. The HVHP-04700 (pore size 0.45 µm and thickness 125 µm, ø: 5.5 cm, DUPAPORE^®^), a hydrophobic porous polyvinylidene fluoride film, was obtained from Millipore Corp. The ionic liquids (ILs) were selected from [C_4_mim][BF_4_] and [C_4_mim][Tf_2_N] produced by the Center for Green Chemistry and Catalysis, Lanzhou Institute of Chemical Physics, Chinese Academy of Sciences. Figure 10 shows the molecular structures of the ionic liquids used in this study.

### 3.2. Preparation of the Ionic-Liquid-Driven Supported Liquid Membrane, the La, Ce, and Tb Supply Phase, and the PO_4_^3^^−^ Supply Phase

To prepare the ionic-liquid-driven supported liquid membrane, the hydrophobic porous polyvinylidene fluoride film (HVHP-04700) was immersed in an ionic liquid (≥200 µL of either [C_4_mim][BF_4_] or [C_4_mim][Tf_2_N]) for more than 2 h. For the La, Ce, and Tb supply phase, a lanthanum sulfate, cerium sulfate, and terbium sulfate mixed solution (or a lanthanum nitrate, cerium nitrate, and terbium nitrate mixed solution) was prepared in a calibrated volumetric flask by dissolving each compound in ultrapure water at a suitable La/Ce/Tb molar ratio. Each compound was placed in an ultrasound cleaner for 20 min to ensure complete dissolution. The PO_4_^3^^−^ supply phase (1 M) was prepared in a calibrated volumetric flask by the dilution of concentrated phosphoric acid.

### 3.3. LaPO_4_:Ce^3+^, Tb^3+^ Precursor Synthetic Process for the Membrane Reaction

The experiment was carried out in a glass cell system with self-adjusting diffusion which consisted of two glass units sandwiching a functional membrane ([C_4_mim][BF_4_] or [C_4_mim][Tf_2_N]). The glass units were filled with 50 mL of the rare earth mixture (rare earth sulfates or rare earth nitrates) at a certain ratio and 50 mL of the phosphoric acid solution (1 M). To ensure a homogeneous system, both solutions were stirred with a magnetic stirrer at 1000 rpm (Mini MR, IKA). After the complete reaction had occurred at room temperature, the white product was collected, centrifuged, and washed with ethanol more than 5 times, and then dried at 60 °C in a drying oven for 12 h to obtain the precursor. Under a set temperature (1000 °C), the LaPO_4_:Ce, Tb precursor powder was calcined for 1 h under a reducing atmosphere to finally produced the green-emitting phosphors.

### 3.4. Characterization Methods

The concentration of P in the ultrapure water phase and the rare earth ion concentration of the LaPO_4_:Ce^3+^, Tb^3+^ samples were measured using a HORIBA-Jobin Yvon ULTIMA 2 series by inductively coupled plasma-atomic emission spectroscopy (ICP-AES). The Raman spectra were measured using the Horiba Jobin Yvon S.A.S. LabRAM Aramis. The structures and phase purities of the as-prepared LaPO_4_:Ce^3+^, Tb^3+^ samples were identified using X-ray diffraction analysis with a Bruker AXS D8 Advance Powder X-ray diffractometer (Cu Kα radiation, λ = 1.5418 Å). The morphologies, energy spectrum of membranes, and the as-prepared products were observed under a ZEISS SIGMA 500 field emission scanning electron microscope. The excitation and emission spectra were taken on an Edinburgh FLS980 spectrometer equipped with a 450 W ozone-free xenon arc lamp as the excitation source.

## 4. Conclusions

In summary, LaPO_4_:Ce, Tb phosphors with monoclinic structures and good photoluminescence were successfully synthesized using a novel, controllable, and efficient ionic-liquid-driven supported liquid membrane system. The release rate of PO_4_^3−^ from the liquid membrane with different ionic liquids was different. The phosphors prepared by this method exhibited micro-spherical (when using rare earth sulfates as the raw material) and nanoscale stone-shape (when using rare earth nitrates as the raw material) morphologies due to the influence of the different anions. These studies indicate that ionic-liquid-driven supported liquid membrane systems are a promising method for preparing LaPO_4_:Ce^3+^, Tb^3+^ phosphors.

## Figures and Tables

**Figure 1 ijms-20-03424-f001:**
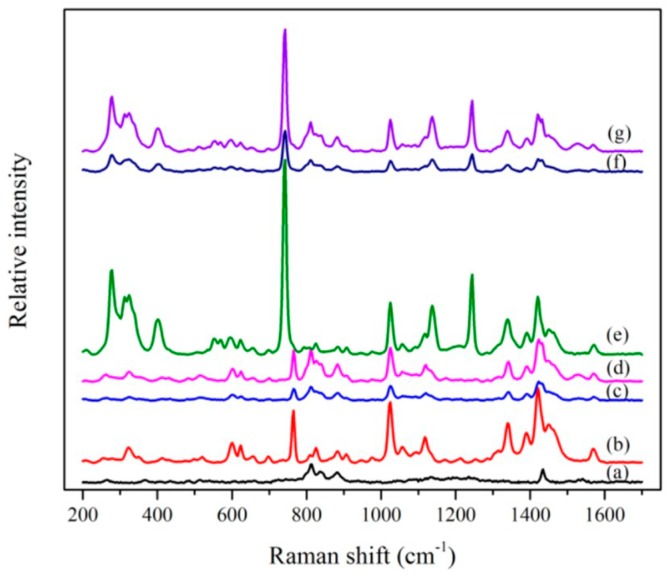
Comparison of Raman spectra of (a) untreated HVHP membrane, (b) [C_4_mim][BF_4_], (c) the surface of the [C_4_mim][BF_4_]-driven HVHP, (d) the internal part (40 µm below the surface) of the [C_4_mim][BF_4_]-driven HVHP, (e) [C_4_mim][Tf_2_N], (f) the surface of the [C_4_mim][Tf_2_N]-driven HVHP membrane and (g) the internal part (40 µm below the surface) of the [C_4_mim][Tf_2_N]-driven HVHP membrane.

**Figure 2 ijms-20-03424-f002:**
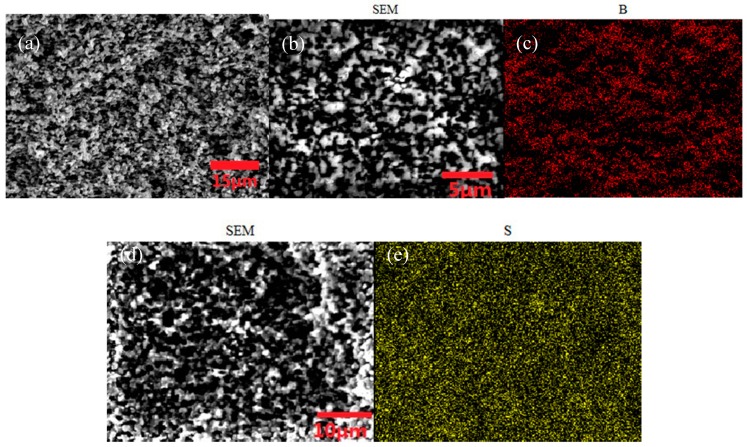
Plot of (**a**) micrographs of the cross-section of the untreated HVHP membrane, (**b**) micrographs of the cross-section of the [C_4_mim][BF_4_]-driven HVHP, (**c**) map microanalysis (B) of [C_4_mim][BF_4_]-driven HVHP, (**d**) micrographs of the cross-section of [C_4_mim][Tf_2_N] and (**e**) map microanalysis (S) of the [C_4_mim][Tf_2_N]-driven HVHP membrane.

**Figure 3 ijms-20-03424-f003:**
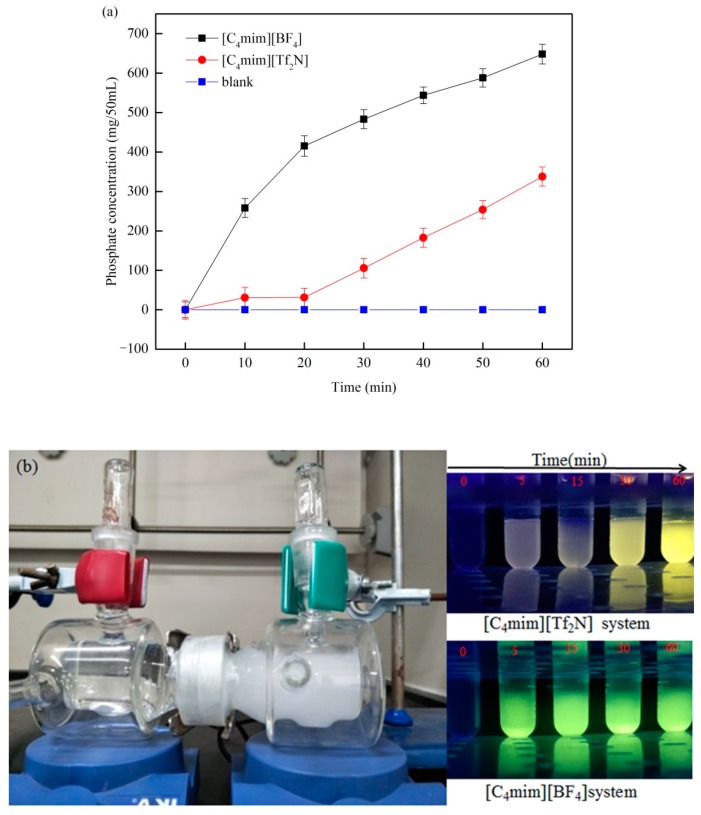
Plot of (**a**) the concentration of PO_4_^3^^−^ that has crossed the liquid membrane within 60 min using different ionic liquids in the liquid membrane phase and (**b**) picture of the phosphor reaction process and images of the precursor solution color within 60 min when using different ionic liquids in the liquid membrane phase.

**Figure 4 ijms-20-03424-f004:**
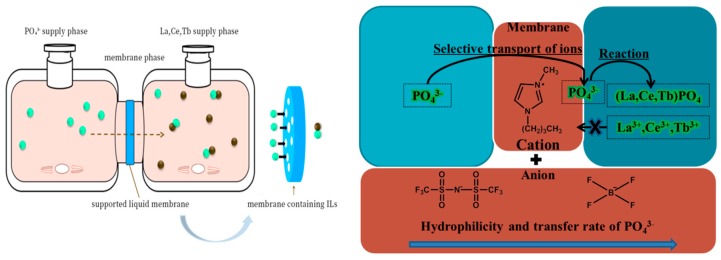
Ionic-liquid-driven supported liquid membrane system and schematic diagram of reaction mechanism.

**Figure 5 ijms-20-03424-f005:**
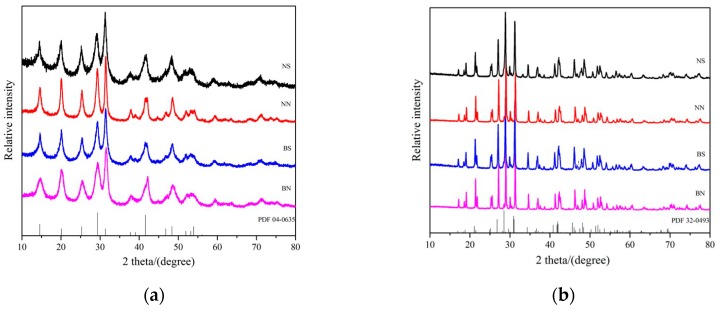
XRD patterns of the precursors (**a**) and calcined LaPO_4_:Ce^3+^, Tb^3+^ phosphors (**b**) prepared under different conditions.

**Figure 6 ijms-20-03424-f006:**
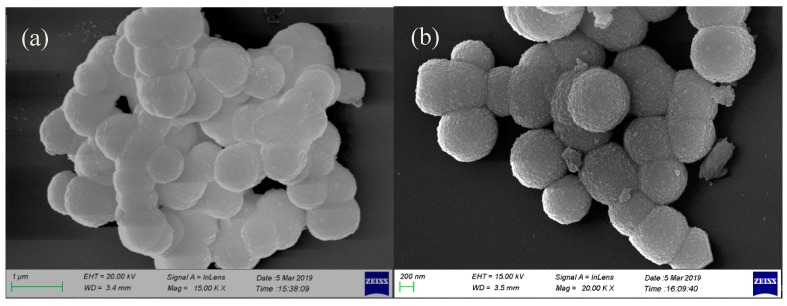
SEM images of the precursors prepared under different conditions: (**a**) BS, (**b**) NS, (**c**) BN, and (**d**) NN. (EHT for extra high tension, WD for working distance)

**Figure 7 ijms-20-03424-f007:**
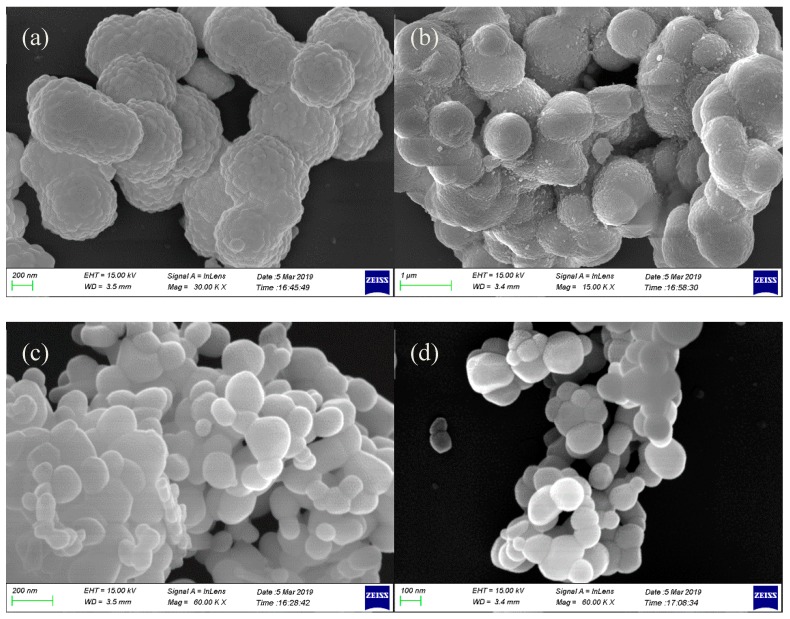
SEM images of calcined LaPO_4_:Ce^3+^, Tb^3+^ phosphors prepared under different conditions: (**a**) BS, (**b**) NS, (**c**) BN, and (**d**) NN. (EHT for extra high tension, WD for working distance)

**Figure 8 ijms-20-03424-f008:**
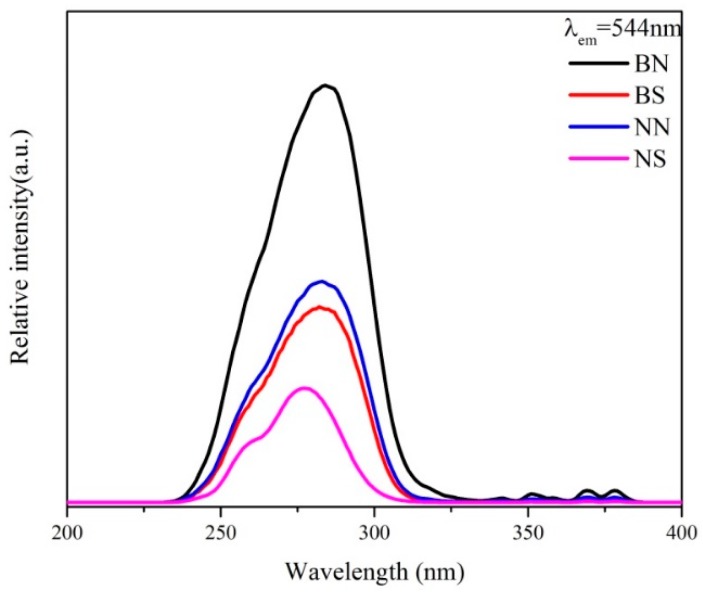
The excitation spectra of the calcined LaPO_4_:Ce^3+^, Tb^3+^ phosphors prepared under different conditions.

**Figure 9 ijms-20-03424-f009:**
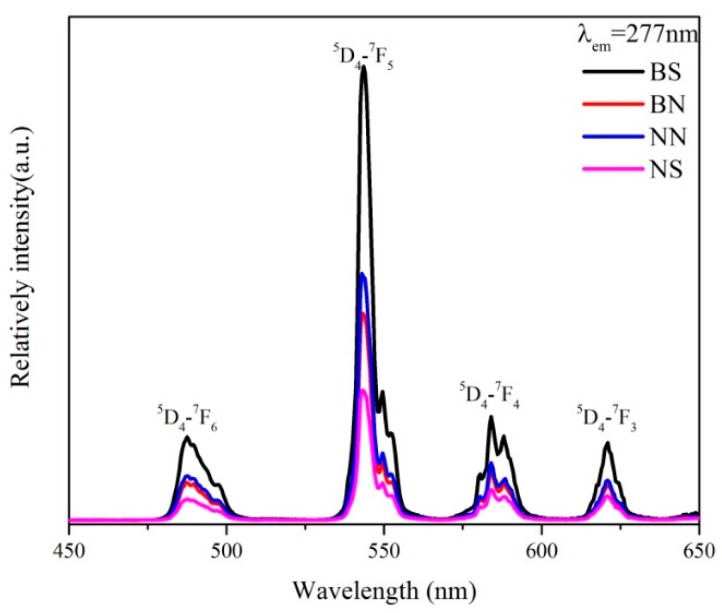
The emission spectra of the calcined LaPO_4_: Ce^3+^, Tb^3+^ phosphors prepared under different conditions.

**Figure 10 ijms-20-03424-f010:**
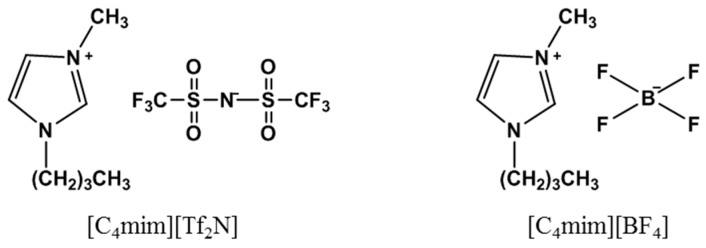
The molecular structures of the different ionic liquids used.

**Table 1 ijms-20-03424-t001:** The molar ratio of a mixed solution of the rare earth elements and the molar ratio of the precursor.

Samples	La^3+^ Molal Percent	Ce^3+^ Molal Percent	Tb^3+^ Molal Percent
Initial solution	55.13	31.04	13.83
Initial precursors	53.40	34.99	11.61
After adjusting solution	55.70	26.20	18.09
After adjusting precursors	55.00	30.11	14.89

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
