# Peer review of "Simple Preparation of LaPO4:Ce, Tb Phosphors by an Ionic-Liquid-Driven Supported Liquid Membrane System"

_ijms, 2019, doi:10.3390/ijms20143424_

Round 1
Reviewer 1 Report
The manuscript entitled: “Simple preparation of LaPO4:Ce, Tb phosphors by an ionic liquid-driven membrane system” submitted to International Journal of Molecular Sciences as Article describes a preparation of a phosphors materials in membrane system using ionic liquid based membrane and two types of precursors: nitrates and sulphates. Authors successfully reported of different microstructure, depending on reaction conditions. The material was characterized using SEM, XRD and absorption-emission spectroscopy. This is the second revision of the manuscript in revised version.
After reviewing the corrected manuscript, I am satisfied with modifications done by Authors. I do recommend this article for publishing in its current form, after including small modification.
When Authors adduce to other work from the literature, namely Polymer 92, 2016, 50-57, it is custumal to use author’s surname. Please correct “Krzysztof A” for “Bogdanowicz” in the manuscript.
Reviewer 2 Report
The Authors have answered the reviewer's criticisms and the paper can be now accepted for publication.
This manuscript is a resubmission of an earlier submission. The following is a list of the peer review reports and author responses from that submission.
Round 1
Reviewer 1 Report
The manuscript entitled: “Simple preparation of LaPO4:Ce, Tb phosphors by an ionic liquid-driven membrane system” submitted to International Journal of Molecular Sciences as Article describes a preparation of a phosphors materials in membrane system using ionic liquid based membrane and two types of precursors: nitrates and sulphates. Authors successfully reported of different microstructure, depending on reaction conditions. The material was characterized using SEM, XRD and absorption-emission spectroscopy.
The topic presented in the manuscript discuss new approach for photoluminescence materials preparation using membrane technology. In my opinion described in manuscript new way of specific and controlled synthesis of phosphorus could be of interest for scientific community. However due to some mayor issues, I do not recommend this article for publishing in its current form.
Here is the list of issues that requires authors attention:
1. Section 2.1: In my opinion the characterization of the ionic liquid-driven HVHPis not sufficient. Authors in this section should provide some explanation why this particular membrane-ionic liquid combination was used. How authors got to the conclusion that the pores of HVHP membrane are entirely filled with ionic liquid? In my opinion presented micrographs alone does not provide sufficient evidence for such statement. I would suggest using other complementary techniques like in Polymer 92 (2016) 50-57.
2. Section 2.1, Fig. 1: Which part of the membrane is presented in the figure?
3. Section 2.2, page 3, line 99, 100: How authors perform the analysis of the concentration inside of the membrane as suggest the description? (concentration of PO43- across the liquid-membrane).
4. Section 2.2, in general: this section is not clear for the reader since in there is no explanation about the experimental setup, experiment description prior to this section.
5. Section 2.2: Did authors check the solubility of ionic liquid in tested media? It is important to check if there are no losses in the transporting media inside the membrane.
6. Section 2.4: In my opinion a description of how samples were prepared (just in brief) is missing in this section.
7. Section 2.6: this section should be placed in previous section, 2.2.
8. Fig. 9 should be place at the beginning of experimental/results section because it provides important overview about the experiment.
Reviewer 2 Report
The Authors present an experimental investigation concerning the preparation of LaPO4 :Ce, Tb phosphors using a ionic-liquid-driven supported polymeric membrane.
The results are of interest to the community working in the field. They are able to obtain phosphors of different shapes and good photoluminescence.
The work is generally well conducted and the paper is clearly written and focussed. Therefore the manuscript may become publishable. However there are a couple of issues that the Authors should carefully addressed in their revised version.
1) the characterization of the ionic-liquid supported membrane. There is not any clear evidence of the pore filling of the membrane by the IL. How can the Authors exclude the formation of just a superficial thin film of IL on the surface rather than a filling of the pores? The Authors report as evidence three SEM figures, saying that from the Figure it is evident that the IL has filled the pores, however I cannot see any evidence of that.
2) the proposed mechanism is quite simple and it does not explain why the two different ILs have such a different performance. The Authors state that the cation is responsible of the phosphate transfer, however both ILs have the same cation, so the role of the anion, which must be significant, is not clear.
3) Finally, together with a comparison between the two ILs the Authors should report the performance of the membrane without ILs, as a reference.